# Poor long-term outcomes despite improved hospital survival for patients with cryptococcal meningitis in rural, Northern Uganda

Mark Okwir [1,2,3]*, Abigail Link[4,5], Bosco Opio[2], Fred Okello[3], Ritah Nakato[2,3], Betty Nabongo[3], Jimmy Alal[3], Joshua Rhein[6,7], David Meya[6,7], Yu Liu[1], Paul R. Bohjanen[1,2,6,7,5]*

1 Department of Public Health Sciences, University of Rochester Medical Center, Rochester, New York, United States of America, 2 Department of Internal Medicine, Faculty of Medicine, Lira University, Lira, Uganda, 3 Department of Medicine, Lira Regional Referral Hospital, Lira, Uganda, 4 School of Nursing, University of Washington, Seattle, Washington, United States of America, 5 Department of Medicine, Division of Infectious Diseases, University of Rochester, Rochester, New York, United States of America, 6 Infectious Diseases Institute, College of Health Sciences, Makerere University, Kampala, Uganda, 7 Division of Infectious Diseases and International Medicine, Department of Medicine, University of Minnesota, Minneapolis, Minnesota, United States of America

* Mark_Okwir@urmc.rochester.edu (MO); Paul_Bohjanen@urmc.rochester.edu (PRB)

## Abstract

### Background

Cryptococcal meningitis (CM) remains a major cause of death among people living with HIV in rural sub-Saharan Africa. We previously reported that a CM diagnosis and treatment program (CM-DTP) improved hospital survival for CM patients in rural, northern Uganda. This study aimed to evaluate the impact on long-term survival.

### Methods

We conducted a retrospective study at Lira Regional Referral Hospital in Uganda evaluating long-term survival ($\geq$1 year) of CM patients diagnosed after CM-DTP initiation (February 2017-September 2021). We compared with a baseline historical group of CM patients before CM-DTP implementation (January 2015-February 2017). Using Cox proportional hazards models, we assessed time-to-death in these groups, adjusting for confounders.

### Results

We identified 318 CM patients, 105 in the Historical Group, and 213 in the CM-DTP Group. The Historical Group had a higher 30-day mortality of 78.5% compared to 42.2% in the CM-DTP Group. The overall survival rate for the CM-DTP group at three years was 25.6%. Attendance at follow-up visits (HR:0.13, 95% CI: [0.03–0.53], $p$ <0.001), ART adherence (HR:0.27, 95% CI: [0.10–0.71], $p$ = 0.008), and fluconazole adherence: (HR:0.03, 95% CI: [0.01–0.13], $p$ <0.001), weight >50kg (HR:0.54, 95% CI: [0.35–0.84], $p$ = 0.006), and performance of therapeutic lumbar punctures (HR:0.42, 95% CI: [0.24–0.71], $p$ = 0.001), were

**Data Availability Statement:** https://doi.org/10.6084/m9.figshare.25447570.v1.

**Funding:** R.B; was supported by NIH grant R21TW012439, the Fulbright U.S. Scholars Program, the University of Minnesota Medical

School, and the University of Rochester Medical Center. A.L.; was supported through the Robert Wood Johnson Foundation Future of Nursing Scholars Fellowship; GO Health Travel Fellowship, University of Washington Department of Global Health; Boeing International Fellowship, University of Washington Graduate School; Hester McClaws Dissertation Fellowship; Fulbright Student Fellowship; NIH Fogarty Global Health Fellowship grant D43TW009345; NIH training grant T32AI055433; and NIH grant R21TW012439. M. O.; was supported by the Fulbright Foreign Scholars Program, the University of Rochester Department of Public Health Sciences, and NIH grant R21TW012439. The funding supported study design, implementation, data collection, analyses, and publication.

**Competing interests:** The authors have declared that no competing interest exist.

associated with lower risk of death. Altered mentation was associated with increased death risk (HR: 1.63, 95% CI: 1.10–2.42, $p$ = 0.016).

## Conclusion

Long-term survival of CM patients improved after the initiation of the CM-DTP. Despite this improved survival, long-term outcomes remained sub-optimal, suggesting that further work is needed to enhance long-term survival.

## Introduction

Cryptococcal meningitis (CM) is a leading cause of death among people living with HIV, accounting for 15% of all AIDS-related deaths globally and 181,100 deaths in sub-Saharan Africa (SSA) in 2014 [1]. In 2020, the estimated global burden of HIV-associated cryptococcal infection accounted for 19% (13–24%) of AIDS-related mortality [2]. Despite increased antiretroviral therapy (ART) use during the "test and treat" era [3], the burden of CM remains unacceptably high in SSA [4, 5]. and accounts for 60% of all meningitis cases in Uganda [6]. Long-term survival of CM patients has been under-studied in routine care settings despite the high burden of CM in SSA [6].

Despite decreased CM prevalence globally, high mortality persists [6, 7]. In rural, northern Uganda, we reported high mortality, with 90.9% of CM patients being ART-experienced at admission [8]. Therefore, it is important to determine why high mortality persists among CM patients despite the improved availability of ART [9, 10]. During the pre-ART era few patients survived long-term [10, 11], as the median survival for CM was 26 days, and 14-day survival was 47% [12]. Currently, the long-term mortality of CM ranges from 13% to 78% globally [13], primarily reported from studies involving routine care or randomized control trials [6].

Risk factors associated with long-term survival from CM after hospital discharge remain largely uncharacterized outside of research settings. Previous studies identified factors linked to short-term survival, and these same factors were found to be predictive of long-term survival outcomes [13–15]. Those factors associated with improved long-term survival of CM patients include effective antifungal therapy use in the induction phase [14] and ART use [13, 14, 15]. Additionally, the performance of therapeutic lumbar punctures (LPs) during hospitalization was also associated with improved long-term survival [16]. Fluconazole as prophylaxis to prevent CM relapse is recommended by the World Health Organization (WHO) and the Ugandan Clinical Guidelines [17, 18] although, when to stop fluconazole in the long term remains unclear. Conversely, factors associated with poor long-term survival of CM patients include high cerebrospinal fluid (CSF) antigen titers, immune reconstitution inflammatory syndrome (IRIS) [19–21], altered mental status, and seizures [22], all of which largely occur during hospitalization. Despite several efforts to improve hospitalization outcomes among CM patients, less effort has been made toward improving outcomes following hospital discharge.

In February of 2017, a standard of care CM diagnosis and treatment program (CM-DTP) began at Lira Regional Referral Hospital (LRRH) in rural, northern Uganda (9) based on the WHO and Uganda Clinical Guidelines [17, 18]. Prior to initiation of the CM-DTP at LRRH, diagnosis, and treatment of CM patients was based on the clinicians' judgment due to a lack of laboratory diagnostic and monitoring capabilities as well as the unavailability of antifungal agents such as Amphotericin B, all of which are common challenges to CM care in non-research centers in SSA [23]. Because of the inaccessibility and high cost of Amphotericin B,

implementation of CM treatment guidelines at LRRH was challenging prior to initiation of the CM-DTP, and outcomes for CM patients were poor with >75% of deaths occurring during hospitalization. This information was the impetus for the CM-DTP at LRRH. Although the current study is focused on rural, northern Uganda, this project provides a true representation of many other rural settings in SSA where resources are limited, to determine if following standard-of-care protocols could lead to improved outcomes. We found that the CM-DTP led to increased diagnosis of CM, improved treatment, and improved hospital outcomes for patients with CM and other forms of meningitis [9]. Here, we report on the impact of this standard-of-care program on the long-term survival of CM patients and the factors associated with long-term survival.

## Methods

### Design, study population, and setting

To assess long-term survival, we conducted a retrospective review of two groups of CM patients. One group included 213 CM patients living with HIV who were hospitalized at LRRH from February 2017 through September 2021 and enrolled in the CM-DTP (CM-DTP Group). This is part of an ongoing Meningitis Treatment Program we previously reported (May 2022) to have improved hospital outcomes [9]. Between September 2022 to January 2023, we accessed the data to evaluate the long-term survival of CM patients [data] [24]. To provide a historical context on CM care prior to the initiation of the CM-DTP, we also evaluated a historical group of 105 CM patients treated at LRRH between February 2015 and February 2017 (Historical Group). In the Historical Group, doctors relied on clinical judgment, and available tests including rapid lateral flow assays for CM diagnosis, gram stain and culture, and blood chemistry for monitoring CM patients were usually not performed because they were unavailable in the hospital or were too expensive at outside commercial laboratories for patients to afford. We included all adult CM patients (aged ≥18 years) with hospital records, within the study periods. We excluded patients without complete hospital records or known hospital dispositions. Patients were considered to have confirmed CM if they had laboratory-confirmed diagnoses of CM. We identified 176 of 213 patients with confirmed CM and 37 patients were presumed CM in the CM-DTP Group. In the Historical Group, 48 out of 105 patients were confirmed to have CM. Additionally, patients were considered to have presumed CM if they were treated for CM based on clinical judgment by the treating doctors. Long-term survival was defined as survival of 12 months or longer from the first day of hospitalization. The CM-DTP Group underwent follow-up throughout hospitalization, with additional data subsequently collected from medical records during scheduled clinic visits at LRRH. Post-discharge follow-up was based on records from follow-up appointments at LRRH and phone calls to patients, family members, and/or other health centers.

This study was conducted at LRRH in northern Uganda, a 350-bed government-owned public hospital located 340 kilometers from Uganda's capital, Kampala [25]. This hospital serves a population of 2.2 million Ugandans and is a high-volume hospital with 232,014 outpatient visits and 12,203 inpatient admissions yearly (as of 2022) [25].

### CM care in the CM-DTP group

Confirmed CM was diagnosed using a cryptococcal antigen (CrAg) lateral flow assay (IMMY, Norman, Oklahoma) on cerebrospinal fluid (CSF) after a lumbar puncture, a CrAg latex agglutination assay, India ink, CSF culture, positive Biofire (PCR-based test) or CSF gram stain showing yeast as previously reported [9].

Patients enrolled in the CM-DTP received standard-of-care CM treatment based on WHO and Ugandan guidelines [17, 18], using deoxycholate amphotericin B and oral fluconazole as described previously [9]. Flucytosine was not used because it was unavailable during this study period [26]. The CM-DTP protocol included: Amphotericin B deoxycholate (0.7–1.0 mg/kg/day) which was administered for 7–10 days, and extended to 14 days for inadequate response. Fluconazole was initiated at 800–1200 mg/day during induction, tapered to 400 mg/day in consolidation, and maintained at 200 mg/day. Each Amphotericin B dose was flanked by 1.5 L normal saline infusions. Magnesium and potassium chloride supplementation were routine components of the CM treatment regimen. The specific doses depended on the severity of the deficiency and other factors like renal function. In addition, pre-, intra-, and post-treatment monitoring included: creatinine, electrolytes, and hemoglobin levels, detailed HIV and ART status at CM diagnosis (CD4 count, viral load, and ART use and adherence, prior CM history). ART adherence was assessed using standard pill counts, patient self-reported adherence, or information from ART cards issued by HIV clinics which also included duration, and type of ART. CM-DTP patients had follow-up at LRRH after discharge where active assessments could be performed during their monthly return visits for drug refills (ART and Fluconazole).

## CM care in the Historical Group (prior CM-DTP)

Patients with CM in the Historical Group received treatment based on the clinician's judgment at the time they presented to the hospital. Supportive treatment was often inadequately provided or in some cases not available. Medical supplies were often lacking, with delays in diagnosis and initiation of treatment being a challenge [23, 27]. In addition, a majority of medical workers had insufficient competence to handle CM patients. These challenges are a common situation in a majority of health centers in Uganda [23, 27]. Antifungal therapy given to patients was not uniform, and induction antifungal therapy was comprised of Amphotericin B, intravenous fluconazole, or oral fluconazole depending on availability and patient's ability to pay. Supplementation with magnesium and potassium chloride was not a routine part of the CM therapy due to the lack of access and availability in this rural resource-poor setting. Patients were sent back to their primary HIV clinics after hospital discharge, although a subset was followed in the HIV Clinic at LRRH.

## Outcomes: All-cause mortality among CM patients (CM-DTP)

The primary outcome in this study was all-cause mortality among CM patients who were treated using standard-of-care (CM-DTP Group). We also assessed the factors associated with long-term survival among CM patients who participated in the CM-DTP, including the impact of adherence to ART and the maintenance phase of fluconazole therapy (secondary prophylaxis). Records of CM deaths were from patient files and the follow-up registry. We also made phone calls to patients, next of kin, and/or their primary HIV clinics if the patients were unreachable. At the analysis time, we inquired if the patient was alive or deceased, physical status, comorbidities, and complications (including neurological deficits such as blindness). If the patient died, we recorded details surrounding their death such as date of death, location of death, and cause.

## Study data and covariates

Covariate data were extracted from hospital records and clinical record forms (CRFs) and included: age, sex, clinical presentations, category of CM diagnoses (new, relapse, or Immune reconstitution Syndrome, symptom duration, CD4, HIV viral load, lumbar punctures, medicines utilized, ART history, ART and fluconazole adherence, laboratory results, diagnoses, CM

management, and outcomes. After hospital discharge, information was obtained from outpatient and follow-up records. Lastly, routine clinic review notes in the patient's medical record notebooks provided additional information on adherence.

### Patient consent statement

This study protocol was approved by the Gulu University Research Ethics Committee (GUREC-066-19), the Mbarara University of Science and Technology Research Ethics Committee (MUSTREC—Reference Number: 1/7), the Uganda National Council of Science and Technology (UNCST: HS 2675), and the University of Minnesota IRB (STUDY00011386). The study met the international human subject protection standards. We sought written informed consent from the patients and no information used could identify them.

### Statistical analyses

Descriptive statistics utilized univariate analysis; categorical variables were compared using the Chi-Square test or Fisher's Exact test. Continuous variables were reported as means (standard deviation), medians (interquartile ranges), and proportions or percentages. We performed Chi-Square tests to describe mortality in the CM-DTP group. We performed multivariate regression analysis using the Cox Proportional Hazard regression model. The outcome variable was survival time ("time-to-death") for the survival status. Missing data on outcome (missing at random, 6.6%) were not included in the analyses. We conducted unadjusted and adjusted Cox regression analyses, modeling the survival time with the covariates. We regressed secondary exposures (Fluconazole adherence, ART status, and follow-up adherence) with the outcome. We accounted for both time-varying factors (such as weight) and time-fixed variables (such as sex) in our regression models. We adjusted for confounding variables using regression models based on the directed acyclic graph (DAG) and compared CM mortality outcomes between groups using Kaplan-Meier analyses. Historical data limitations (non-comparative treatment, poor 60-day survival [<15%], low long-term survival, and poor follow-up information) restricted analyses to descriptive statistics for the Historical Group. We reported effect estimates and risk (hazard ratio) with 95% CI at 0.05 two-sided alpha level of significance. We used Stata version 17 (StataCorp, College Station, TX, USA), and R statistical software (Version 4.2.3, Vienna, Austria, The R Foundation of Statistical Computing) for data analyses.

## Results

### Characteristics of the study population

We identified 318 CM patients: 105 patients in the Historical Group, and 213 patients in the CM-DTP Group. The majority were male (54.7%) with a median age of 35 years (IQR: 30–44), and the median weight was 53 Kg (IQR: 46–58.0). (Table 1). All patients had confirmed HIV infection, 31 (11%) were ART naïve, and 217 (77.2%) patients were on ART and considered to be adherent. We found a significant difference in the proportion of ART experience between Historical Group (22.3% were ART naïve) and the CM-DTP Group (4.5% were ART naïve). Most patients, 247/291 (84.9%) presented with an index episode of CM. The median CD4 count was 63 cells/mm$^3$ (IQR: 22–149) and the median HIV viral load was 3.9 x 10$^4$ copies/mL (IQR: 0.4–15.5). CD4 counts were performed in only 104 of 318 (32.7%) patients and viral loads were performed in 28 of 318 (8.8%) patients, primarily because of lack of availability. Most CM-DTP Group patients, 181 of 213 (85%) underwent LP compared to 43 of 105 (41%) patients in the Historical Group.

**Table 1. Characteristics of the study population.**

| | ALL Patients | Historical Group | CM-DTP Group |
|---|---|---|---|
| | N = 318 | n = 105 | n = 213 |
| **Clinical Characteristics** | | | |
| Gender, females, n (%) | 144 (45.3) | 40 (38.1) | 104 (48.8) |
| Median age (IQR) [b], years | 35 (30–44) | 35 (29–44) | 36 (30–44) |
| Median weight (IQR) [b], Kg | 53 (46–58) | 49 (29–58) | 53 (46–58) |
| CM Diagnosis category, n (%) | 291 (91.5) [b] | 90 (30.9) | 201 (69.1) |
| New, n (%) | 247 (84.9) | 83 (92.2) | 164 (81.6) |
| IRIS, n (%) | 23 (7.9) | 7 (7.8) | 16 (8.0) |
| Relapse, n (%) | 21 (7.2) | 0 | 21 (10.4) |
| TB treatment current, yes, n (%) [b] | 34 (37.0) | 2 (18.2) | 32 (39.5) |
| History of TB, yes, n (%) [b] | 23 (34.3) | 3 (42.9) | 20 (33.3) |
| CM Symptom duration, n (%) | 280 (88.1) [b] | 89 (31.8) | 191 (68.2) |
| Median (IQR) days | 14 (7–28) | 7 (5–21) | 14 (7–29) |
| <14 days, n (%) | 105 (37.5) | 46 (51.7) | 59 (30.9) |
| ≥14 days, n (%) | 175 (62.5) | 43 (48.3) | 132 (69.1) |
| Altered mentation, yes, n (%) | 119 (48.8) | 46 (38.7) | 73 (61.3) |
| **Laboratory Characteristics** | | | |
| CD4 cell count, n (%) | 104 (32.7) [b] | 41 (39.4) | 63 (60.6) |
| Median CD4 (IQR) cells/mm$^3$ | 63 (22,149) | 56 (15–98) | 74 (31–193) |
| <200 cells/mm$^3$ | 86 (82.7) | 39 (95.1) | 47 (74.6) |
| ≥200 cells/mm$^3$ | 18 (17.3) | 2 (4.9) | 16 (25.4) |
| HIV Viral Load, n (%) | 28 (8.8) | 5 (17.9) | 23 (82.1) |
| Median VL (IQR) X10$^4$ copies/mL | 3.9 (0.4–15.5) | 3.2 (0.2–50) | 4.2 (0.5–11.7) |
| **Lumbar punctures done, n (%)** | 224 (70.4) [b] | 43 (41.0) | 181 (85.0) |
| 0–1, n (%) | 92 (41.1) | 35 (81.4) | 57 (31.5) |
| 2–3, n (%) | 100 (44.6) | 7 (16.3) | 93 (51.4) |
| ≥4, n (%) | 32 (14.3) | 1 (2.3) | 31 (17.1) |
| Average (Min, Max) | 2.1 (0,5) | 1.2 (0,5) | 2.4 (0,5) |
| **ART status, n (%)** | 281 (88.4) [b] | 103 (36.7) | 178 (63.3) |
| ART Naive | 31 (11.0) | 23 (22.3) | 8 (4.5) |
| ART uninterrupted | 217 (77.2) | 77 (74.8) | 140 (78.7) |
| Defaulted | 12 (4.3) | 2 (1.9) | 10 (5.6) |
| Stopped but restarted | 21 (7.5) | 1 (1.0) | 20 (11.2) |
| **Mortality, n (%)** | 250 (78.6) | 65 | 185 |
| At 30 days | 129 (51.6) | 51 (78.5) | 78 (42.2) |
| ≤ 60 days | 156 (62.4) | 56 (86.2) | 100 (54.1) |
| **Unable to be traced, n (%)** | 54 (16.9) | 40 (38.1) | 14 (6.6) |

SD—Standard Deviation, CD—cluster of differentiation, IQR -Interquartile Range, CSF–Cerebrospinal
fluid, TB—Tuberculosis, IRIS—immune reconstitution inflammatory syndrome, VL—Viral Load.
() [b] the number of patients with outcomes, is different from the total number of patients due to missing data.

We assessed how often fluconazole was taken by patients following hospital discharge. Although fluconazole treatment was recommended for all patients after hospital discharge, the majority in the Historical Group, 49/54 (90.7%) had no record of continued fluconazole use following discharge compared to 94/207 (45.4%) in the CM-DTP Group.

## Survival of CM patients in the Historical Group was poor

Patients in the Historical Group did not receive uniform CM care before the CM-DTP implementation and, most (78.5%) died within 30 days of hospitalization. (Table 1). The 60-day survival rate remained poor at 13.8%. Further long-term survival analyses for this Historical Group were not feasible because 60-day survival was less than 15%, non-comparable treatments were given, and lack of long-term follow-up information, with 38.1% of patients not traceable. Nevertheless, the poor 60-day survival in the Historical Group provided background information to suggest that the CM-DTP did improve long-term survival as described below.

## Long-term survival improved in the CM-DTP group

We assessed the long-term survival of patients in the CM-DTP Group using univariate Chi-Square tests and Kaplan-Meier survival curves (Table 2 and Fig 1). Survival status was available for 199 of 213 patients (93.4%) from hospitalization through the follow-up period after hospital discharge. Patients who could not be traced were lost to follow-up and were not included in survival analyses (14/213, 6.6% of CM patients). Overall, the follow-up person-days contributed by 199 patients was 37,876 days (103.8-person years), and a mortality rate of 143 deaths/100-person years. The proportion of CM deaths at 3 years was 148/199 (74.4%). (Table 2). More females (80.2%) died compared to males (69.4%), but no age differences were found in the proportion of CM deaths. The overall long-term survival among all patients was poor, with of 25.6% survival rate at three years. The mean survival time was 256 days (SD: 401) and the median was 39 days (IQR: 11–367). Thus, although in-hospital mortality improved after initiation of the CM-DTP, [9] long-term survival remained poor.

## ART Adherence and prolonged fluconazole use improved long-term survival

Follow-up care and adherence to fluconazole, ART, and follow-up clinic attendance were evaluated to determine the association with the long-term survival of patients (Table 3). We found that patients who adhered to treatment guidelines had better survival compared to those who did not. Those who adhered to fluconazole after hospital discharge had a 98% decreased hazard of mortality compared to those who did not continue fluconazole (HR: 0.02, 95% CI: (0.01, 0.04), $p$-value <0.001). (Table 3). In addition, patients who adhered to ART as per the Ugandan guidelines prior to hospital admission had better survival than those who did not. Patients who adhered to ART had a 65% lower relative hazard of mortality compared to ART naïve patients, (HR: 0.35, 95% CI: (0.21, 0.59) $p$-value <0.001). This beneficial effect of ART adherence remained at 41% in the adjusted analyses (HR: 0.59, 95% CI: (0.34, 1.01), $p$-value 0.054). (Table 3). Patients who adhered to routine outpatient follow-up visits had a 0.02 relative hazard of mortality compared to those who had no follow-up, (HR: 0.02, 95% CI: (0.01, 0.05), $p$-value <0.001). Survival benefits were observed, even among those with incomplete follow-up, compared to those who never participated in follow-up, (HR: 0.12, 95% CI: (0.08, 0.18), $p$-value <0.001). (Table 3).

## Predictors of long-term survival for the CM-DTP group

We conducted a subgroup analysis of patients to identify additional predictors of long-term survival in those who participated in the CM-DTP. Adherence to fluconazole predicted favorable long-term survival among patients (Table 4). Those who adhered had a 97% decreased risk of mortality compared to those who did not take maintenance fluconazole (HR: 0.03, 95% CI: (0.01, 0.13), $p$-value <0.001). In addition, patients who adhered to ART before

**Table 2. Mortality of CM patients in the CM-DTP group.**

| ALL CM patients | | Numbers N = 213 n [d] | Deaths n (%) | Chi-Squared, $X^2$ P-value |
|---|---|---|---|---|
| **Gender** | | 199 | 148 (74.4) | 0.083 |
| | **Female** | 91 | 73 (80.2) | |
| | **Male** | 108 | 75 (69.4) | |
| **Age, years** | | 199 | 148 (74.4) | 0.632 |
| | ≤35 years | 86 | 62 (72.1) | |
| | >35 years | 113 | 86 (76.1) | |
| **Altered mentation** | | 166 | | |
| | No | 104 | 75 (72.1) | 0.420 |
| | Yes | 62 | 49 (79.0) | |
| **ART status** | | 160 | 116 (72.5) | 0.281 |
| | ART Naïve | 6 | 6 (100) | |
| | Adherent | 125 | 89 (71.2) | |
| | Defaulted | 9 | 5 (55.6) | |
| | Restarted | 20 | 16 (80.0) | |
| **Weight, kgs** | | 128 | 98 (76.6) | **0.011** |
| | < = 50 kg | 51 | 45 (88.2) | |
| | >50 kg | 77 | 53 (68.8) | |
| **Fluconazole adherence** | | 192 | 144 (73.4) | **<0.001** |
| | Not used | 92 | 92 (100) | |
| | Adhered | 51 | 9 (17.7) | |
| | Defaulted | 49 | 40 (81.6) | |
| **History of CM** | | 181 | 136(75.1) | 0.216 |
| | Index | 146 | 113(77.4) | |
| | IRIS | 14 | 8(57.1) | |
| | Relapse | 21 | 15(71.4) | |
| **Follow-up adherence** | | 197 | 148 (74.4) | **<0.001** |
| | No follow-up returns | 92 | 92 (100) | |
| | Adhered | 51 | 9 (17.6) | |
| | Defaulted | 49 | 40 (81.6) | |
| **CD4 Category** | | 58 | 41 (70.7) | 0.944 |
| | <200 cells/mm$^3$ | 44 | 31 (70.5) | |
| | ≥200 cells/mm$^3$ | 14 | 10 (71.4) | |
| **Lumbar Punctures** | | 168 | 126 (75.0) | 0.080 |
| | 0–1 | 49 | 42 (85.7) | |
| | 2–3 | 88 | 64 (72.7) | |
| | ≥4 | 31 | 20 (64.5) | |

[d] Numbers of patients with outcomes excluding missing values,

IRIS–Immune Reconstitution Inflammatory Syndrome, Kg–Kilogram(s), $X^2$ –Chi-Squared

hospitalization had a 73% reduced risk of mortality compared to ART-naïve CM patients, (HR: 0.27, 95% CI: (0.10, 0.71), p-value 0.008). Patients who attended follow-up return visits had an 87% reduced risk of mortality compared to those who did not attend, (HR: 0.13, 95% CI: (0.03–0.53), p-value <0.001). Patients who had a weight greater than 50 kgs had a 46% decreased risk of death compared to those less than 50 kgs, (HR: 0.54, 95% CI: (0.38, 0.86), p-value 0.006) suggesting survival benefits in those with better nutritional status. When we evaluated the long-term survival benefits of in-hospital therapeutic LPs among patients, reduced

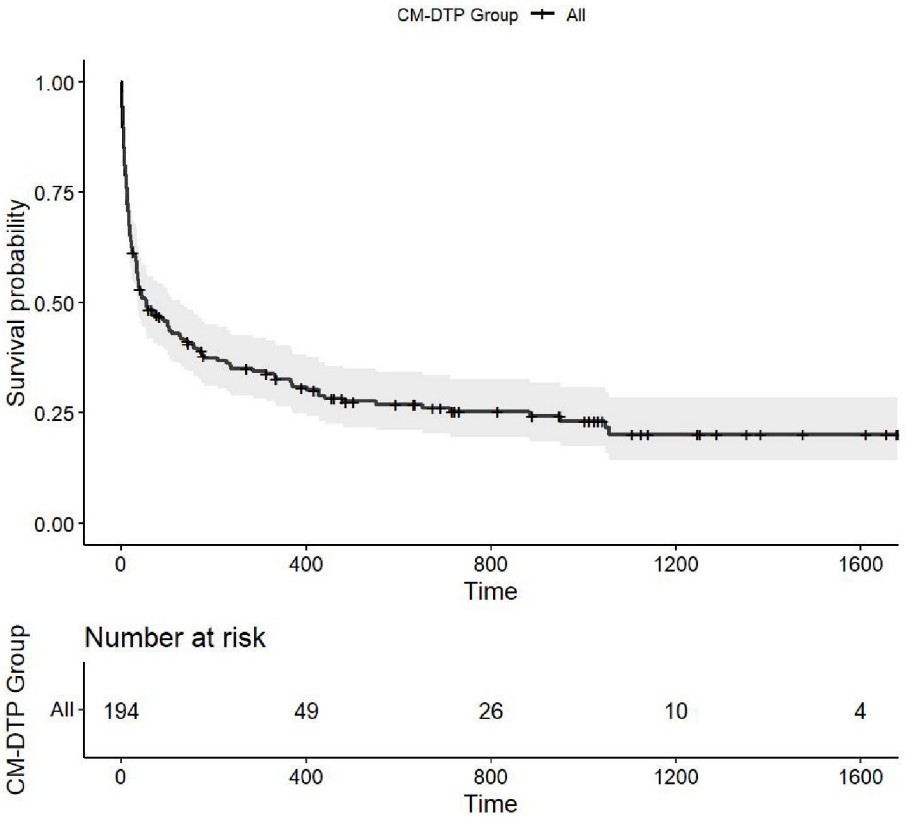

**Fig 1. Survival probability over time (days) and number at risk for CM-DTP group.** The cumulative survival probability curve for patients in the CM-DTP group expressed as a ratio (y-axis) from admission (day 0) is shown over time in days (x-axis).

long-term mortality was observed in those who had 4 or more therapeutic LPs (HR: 0.23, 95% CI: (0.09, 0.60), *p*-value of 0.003) or those who had 2 to 3 therapeutic LPs (HR: 0.46, 95% CI: (0.24–0.88), *p*-value 0.020) compared to those who had less than 2 LPs. Lastly, CM patients who experienced altered mentation during their initial hospitalization had 1.63 times the risk of long-term mortality compared to those without altered mentation (HR: 1.63, 95% CI: 1.10–2.42, *p*-value = 0.016). This suggests that CM patients who had altered mentation possibly had more severe disease which increased the risk of subsequent death post-hospitalization.

## Discussion

We observed long-term survival of CM patients improved following the implementation of the standard-of-care (CM-DTP) for CM in rural Uganda. However, post-discharge care including attendance at follow-up visits and adherence to fluconazole prophylaxis and ART were often sub-optimal. Thus, implementation of this program during hospitalization had long-term benefits in a real-world resources-limited setting, but further work is needed to improve CM care after hospital discharge. Although we previously showed that implementation of this CM-DTP improved diagnosis, treatment, and in-hospital mortality among CM patients, [9] the current results show that long-term three-year survival remained poor at only 25.6%, suggesting that large numbers of patients die after hospital discharge. There was no

**Table 3. Crude and an adjusted Hazard Ratios (HR) for association between long-term survival of CM patients and exposures at Cox proportional hazard regression.**

| Variable | CM Total (Deaths) | Crude Hazard Ratios | | | Adjusted Hazard Ratios | | |
|---|---|---|---|---|---|---|---|
| | | HR | 95% CI | *P*-value | HR[v] | 95% CI | *P*-value |
| **Primary Exposure** | | | | | | | |
| Overall Group [w] | | | | | | | |
| **CM-DTP** | 199 (148) | 0.36 | (0.26, 0.48) | <0.001 | 0.35 | (0.24, 0.49) | **<0.001** |
| **Secondary Exposures** | | | | | | | |
| **Fluconazole Adherence [x]** | | | | | | | |
| Not Used | 139 (139) | 1.00 | | | 1.00 | | |
| Adhered | 52 (10) | 0.02 | (0.01, 0.04) | <0.001 | 0.02 | (0.01, 0.04) | **<0.001** |
| Defaulted | 52 (42) | 0.13 | (0.09, 0.20) | <0.001 | 0.13 | (0.08, 0.20) | **<0.001** |
| **ART Status [y]** | | | | | | | |
| Naïve | 17 (17) | 1.00 | | | 1.00 | | |
| Adhered | 174 (137) | 0.35 | (0.21, 0.59) | <0.001 | 0.59 | (0.34, 1.01) | 0.054 |
| Defaulted | 10 (6) | 0.25 | (0.10, 0.64) | 0.004 | 0.59 | (0.22, 1.58) | 0.291 |
| Restarted | 21 (17) | 0.27 | (0.14, 0.54) | <0.001 | 0.50 | (0.25, 1.03) | 0.061 |
| **Follow-up Adherence [z]** | | | | | | | |
| No follow-up returns | 146 (146) | 1.00 | | | 1.00 | | |
| Adhered | 53 (13) | 0.02 | (0.01, 0.04) | <0.001 | 0.02 | (0.01, 0.05) | **<0.001** |
| Defaulted | 62 (50) | 0.12 | (0.08, 0.17) | <0.001 | 0.12 | (0.08, 0.18) | **<0.001** |

[v] Adjusted Hazard Ratio (HR), Confidence Interval (CI); Lumbar puncture (LP); *P* value level of significance <0.05 (indicating significant p values.

[w] Treatment Group adjusted for Age, Sex, and Antiretroviral use status

[x] Fluconazole adherence adjusted for Age, Sex, and Treatment Group

[y] Antiretroviral use status adjusted for Age, Sex, and Treatment Group

[z] Follow-up adherence adjusted for Age, Sex, and Treatment Group

difference in mortality between males and females, similar to an earlier study conducted in Uganda [28]. Even in the interventional CryptoDex trial in Uganda and Asia, a similar trend of sub-optimal long-term survival was observed, with mortality of 60% at 2 years among HIV patients with CM [29, 30]. Conversely, an earlier study in Uganda observed a 5-year survival rate of 42% among 189 CM patients which is higher compared to our study [11]. A systematic review report on HIV-associated meningitis in sub-Saharan Africa reported long-term mortality (6 months) among 614 cases of CM patients of 44% (95% CI: 36%, 52%, $I^2$ = 71%) [6]. However, these studies were conducted in research settings, whereas our study was conducted in a real-world rural setting where healthcare resources are limited. Overall, our study and others suggest that long-term outcomes for CM remain sub-optimal.

Factors associated with improved long-term survival included adherence to fluconazole and ART, attendance at follow-up return visits after hospital discharge, weight above 50Kgs, and receiving >2 LPs during hospitalization, all of which are consistent with systematic review reports [7, 6]. Inadequate follow-up has been reported to be responsible for the sub-optimal long-term outcomes in similar studies in resource-limited settings [31, 32]. Our results suggest, however, that follow-up visit attendance and adherence to ART and fluconazole in the outpatient setting remained sub-optimal even after initiation of the CM-DTP, perhaps because the CM-DTP focused primarily on improving treatment in the inpatient setting. In addition, high fluconazole resistance (24.1%) in relapse isolates might contribute to long-term mortality, despite the absent of routine testing [33]. Further work needs to be done to improve care after hospital discharge.

**Table 4. Predictors of long-term survival of CM-DTP group patients at Cox proportional hazard regression at 3 years.**

| Variable | Category | CM Total (Deaths) | Adjusted Hazard Ratios | | |
|---|---|---|---|---|---|
| | | | HR [f] | 95% CI | P-Value |
| **Sex** | Males | 91 (73) | 1.00 | | |
| | Females | 108 (75) | 1.15 | (0.67, 1.95) | 0.617 |
| **Age** | < = 35 years | 86 (62) | 1.00 | | |
| | > 35 years | 113 (86) | 1.02 | (0.63, 1.65) | 0.926 |
| **Fluconazole Adherence [g]** | Not Used | 92 (92) | 1.00 | | |
| | Adhered | 51 (51) | 0.03 | (0.01, 0.13) | **<0.001** |
| | Defaulted | 49 (40) | 0.68 | (0.18, 2.55) | 0.565 |
| **ART Status [h]** | Naïve | 6 (6) | 1.00 | | |
| | Adhered | 125 (89) | 0.27 | (0.10, 0.71) | **0.008** |
| | Defaulted | 9 (5) | 0.41 | (0.09, 1.78) | 0.233 |
| | Restarted | 20 (16) | 0.44 | (0.15, 1.36) | 0.155 |
| **Follow-up Adherence [k]** | No follow-up return | 92 (92) | 1.00 | | |
| | Adhered | 51 (9) | 0.13 | (0.03, 0.53) | **<0.001** |
| | Defaulted | 49 (40) | 0.05 | (0.01, 0.19) | **<0.001** |
| **Weight Category [m]** | < = 50Kgs | 51 (45) | 1.00 | | |
| | > 50Kgs | 77 (53) | 0.54 | (0.35, 0.84) | **0.006** |
| **Altered mentation [p]** | No | 104 (75) | 1.00 | | |
| | Yes | 62 (49) | 1.63 | (1.10, 2.42) | **0.016** |
| **Lumbar Punctures [m]** | 0–1 LP | 49 (42) | 1.00 | | |
| | 2–3 LPs | 88 (64) | 0.65 | (0.44, 0.97) | **0.033** |
| | > = 4 LPs | 31 (20) | 0.42 | (0.24, 0.71) | **0.001** |

HR [f], Hazard Ratio

[g] Adjusted for sex, age, ART status, follow-up adherence, weight, and LP

[h] Adjusted for sex, age, fluconazole adherence, follow-up adherence, weight, and LP

[k] Adjusted for sex, age, ART status, fluconazole adherence, weight, and LP

[p] Adjusted for Sex, Age, and lumber punctures

[m] Adjusted for sex and Age

The World Health Organization recommends that ART-experienced people who develop CM be evaluated for possible underlying ART treatment failure, potentially through HIV VL tests [26]. However, the challenge of accessing VL tests and ART resistance tests hampers the implementation of this recommendation in resource-limited settings [34], and most of the patients in this study did not receive testing because CD4 counts and viral load tests, which were not routinely available outside of centralized government laboratory processing available only at authorized HIV clinics. Increasing the utilization of these tests could identify patients with HIV treatment failure, and prompt reflex cryptococcal antigen screening for early diagnosis, intervention [34, 35], and possibly mortality benefits [36]. However, no overall survival benefits were observed through CrAg screening [37], except for patients with CrAg titers ≥1:160, who had 2.6-fold higher 6-month mortality compared to patients with titers <1:160 [35]. Other laboratory tests such as quantitative CSF yeast count and clearance, CSF cytokines, CSF white cell count, and antifungal susceptibility have been reported in several studies to affect hospital outcomes [38–41]. These factors were not evaluated in this study because they were not routinely available for clinical care in this setting. Previously, a study from Uganda observed that therapeutic LPs were associated with a 69% relative improvement in survival, regardless of initial intracranial pressure among admitted CM patients [16]. Similarly, we

observed long-term risk reduction of death in those who underwent more LPs among CM patients. This highlights the need for more frequent LPs during induction therapy, to optimize long-term survival.

Limitations to this study included the retrospective design and missing data because we were unable to contact some patients after discharge. Incomplete medication refill records and use of self-reported adherence likely resulted in inaccurate adherence evaluations and recall bias. This could have led to random misclassification of the exposure, biasing the effect estimates towards the null, causing an underestimation of the association. Also, VL testing could not be used to assess the validity of self-reported adherence because of the lack of availability. The Historical Group had a small sample size, and a greater number of patients without records because most of these patients were referred to their primary health units for follow-up care. Therefore, a complete assessment of post-hospitalization deaths was not feasible. Available outcome data for the Historical Group, however, demonstrated evidence of very high short-term mortality. Although the CM-DTP results clearly showed that attendance at clinic visits and medication adherence were factors associated with long-term survival, survivorship bias could have been possible because patients who had survived post-admission could have had factors (residual confounders) that made them survive up to the point in time where long-term follow-up was possible. Therefore, the survival benefits should be interpreted with caution and in the context of this resource-limited setting outside research centers. However, it is logical that attendance at follow-up visits and adherence to medications would be beneficial to survival.

## Conclusion

Our findings suggest that the long-term survival of CM patients is achievable and can be improved through appropriate standard-of-care treatment and follow-up, even in resource-limited settings. Routine follow-up of CM patients (like all other hospitalized medical patients) is often done passively based on the patient's willingness to return, creating a situation that is sub-optimal for many CM patients because they do not return for follow-up care. We observed outpatient follow-up after hospital discharge and medication adherence are key components in CM care to improve long-term survival. During hospitalization, performing therapeutic LP improves long-term survival. In addition, identifying patients with risk factors such as altered mental status or low weight during hospitalization may allow closer monitoring or efforts to improve nutritional status, which could lead to improved outcomes. Our results highlight the need for improved follow-up care after hospital discharge to improve the long-term survival of patients with CM in rural SSA. Despite improved hospital survival, long-term outcomes remained sub-optimal, suggesting that further work is needed to improve follow-up care and enhance long-term survival.

## Acknowledgments

We would like to acknowledge the support received from various institutions in conducting this study. We are grateful to Lira Regional Referral Hospital and the Lira University Faculty of Medicine for their collaborative efforts. We also thank the Infectious Diseases Institute at Makerere University for administrative support and the Department of Public Health Sciences at the University of Rochester for technical support in the publication process.

## Author Contributions

**Conceptualization:** Mark Okwir, Abigail Link, Betty Nabongo, Joshua Rhein, David Meya, Paul R. Bohjanen.

**Data curation:** Mark Okwir, Abigail Link, Bosco Opio, Fred Okello, Ritah Nakato, Betty Nabongo, Jimmy Alal, David Meya, Paul R. Bohjanen.

**Formal analysis:** Mark Okwir, Abigail Link, Bosco Opio, David Meya, Yu Liu, Paul R. Bohjanen.

**Funding acquisition:** Mark Okwir, Abigail Link, David Meya, Paul R. Bohjanen.

**Investigation:** Mark Okwir, Abigail Link, Fred Okello, Ritah Nakato, Jimmy Alal, David Meya, Paul R. Bohjanen.

**Methodology:** Mark Okwir, Abigail Link, Bosco Opio, Fred Okello, Ritah Nakato, Betty Nabongo, Jimmy Alal, David Meya, Yu Liu, Paul R. Bohjanen.

**Project administration:** Mark Okwir, Abigail Link, Ritah Nakato, Joshua Rhein, David Meya, Paul R. Bohjanen.

**Resources:** Mark Okwir, Abigail Link, David Meya, Paul R. Bohjanen.

**Software:** Bosco Opio, Yu Liu.

**Supervision:** Mark Okwir, Abigail Link, Ritah Nakato, Betty Nabongo, Jimmy Alal, David Meya, Paul R. Bohjanen.

**Validation:** Paul R. Bohjanen.

**Writing – original draft:** Mark Okwir, David Meya, Paul R. Bohjanen.

**Writing – review & editing:** Mark Okwir, Abigail Link, Bosco Opio, Ritah Nakato, Betty Nabongo, Jimmy Alal, Joshua Rhein, David Meya, Yu Liu, Paul R. Bohjanen.

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
