## [Decision Letter · Decision Letter 0]

23 Apr 2024

PONE-D-24-11811Poor Long-Term Outcomes Despite Improved Hospital Survival for Patients with Cryptococcal Meningitis in Rural, Northern UgandaPLOS ONE

Dear Dr. Okwir,

Thank you for submitting your manuscript to PLOS ONE. After careful consideration, we feel that it has merit but does not fully meet PLOS ONE’s publication criteria as it currently stands. Therefore, we invite you to submit a revised version of the manuscript that addresses the points raised during the review process.

We look forward to receiving your revised manuscript.

Kind regards,

Felix Bongomin, MB ChB, MSc, MMed, FECMM

Academic Editor

PLOS ONE

Journal Requirements:

Additional Editor Comments:

Kindly address the reviewer's comments.

Reviewers' comments:

Reviewer's Responses to Questions

**Comments to the Author**

1. Is the manuscript technically sound, and do the data support the conclusions?

Reviewer #1: Yes

2. Has the statistical analysis been performed appropriately and rigorously? 

Reviewer #1: Yes

3. Have the authors made all data underlying the findings in their manuscript fully available?

Reviewer #1: Yes

4. Is the manuscript presented in an intelligible fashion and written in standard English?

Reviewer #1: Yes

5. Review Comments to the Author

Reviewer #1: Introduction

1. Line 71: Please add references.

2. The statement is not clear; you mention short-term survival but all the factors mentioned following this are all talking about long-term survival.

3. Much as death and survival are two sides of the same coin, if you mention factors related to death then that’s what you should reference. Otherwise line 69 talks about risk factors of death but the paragraph then goes on to talk about survival. Clarify this.

4. Line 82: Clarify on what this standard of care program entails. It Is unclear in the entire paragraph

Methods

1. Line 108: You have mentioned the tests that were not routinely available, can you mention which tests were being used and the ones you are referring to in line 112? This is not clear in your text.

2. Line 106: You mention 103 patients in historical group but line 114 mentions 105??

Results

1. Verify number of patients in historical group

2. Line 211: Origin of this denominator 330 is not clear.

3. Table 1: The last row of patients who were untraceable; can you provide more information on these. Are they a part of the missing data for all the different variables?

4. Line 228: This is supposed to be reporting results from your study. Factors should be discussed in the discussion.

5. Table 2: Last row; what comparison is this p-value related to?

6. Table 3 results: Which is your outcome measure? Is it a risk or a rate? Results just mention hazard from lines 255-264.

7. Table 3: if you are looking at fluconazole adherence, shouldn’t the reference group be those who adhered? Same comment for follow-up adherence in both tables 3 and 4.

8. Table 3: Cox regression; which variables are being adjusted for? Separate variables are mentioned below the table which are not shown in the model

9. Is it possible that survival was impacted by other comorbidities in this patient group? These have not been mentioned. As you analyze the mortality data as all-cause, it might be better to actually categorize CM associated deaths compared to non-CM. Otherwise, improving outcomes with the suggested interventions would not help if cause of death is non-CM.

6. PLOS authors have the option to publish the peer review history of their article (what does this mean?). If published, this will include your full peer review and any attached files.

Reviewer #1: **Yes: **Sara Nsibirwa

---

## [Author Response · Author response to Decision Letter 0]

29 Apr 2024

PONE-D-24-11811

Poor Long-Term Outcomes Despite Improved Hospital Survival for Patients with Cryptococcal Meningitis in Rural, Northern Uganda

PLOS ONE

Response: Submission done in compliance to the above

Response: Resubmission has a MARKED copy 

Response: Clean Copy (Unmarked) attached

RESPONSE TO THE REVIEWERS

We thank the reviewers for the favorable review of our initial submission and their detailed critique. We have taken note of each concern raised by the reviewers and are submitting an improved manuscript. Here we give point by point rebuttal and clarifications. Technical recommendations that required our attention are all considered, and the changes were made in the resubmitted manuscript (Version, V3).

1. Line 71: Please add references.

Response: reference added 

(page 3)

2. The statement is not clear; you mention short-term survival, but all the factors mentioned following this are all talking about long-term survival.

Response: Sentence in line 71 was paraphrased for clarity.

Previous sentence: Prior studies demonstrated factors associated with short term survival predicted long-term survival.

New Sentence: Previous studies identified factors linked to short-term survival, and these same factors were found to be predictive of long-term survival outcomes.

(page 3)

3. Much as death and survival are two sides of the same coin, if you mention factors related to death then that’s what you should reference. Otherwise, line 69 talks about risk factors of death but the paragraph then goes on to talk about survival. Clarify this.

Response: 

This was corrected to match the sentence flow, clarity and context as mentioned in response 2 above. 

(Page 3)

4. Line 82: Clarify on what this standard of care program entails. It Is unclear in the entire paragraph.

Response: The standard of care program, referred to as the cryptococcal meningitis diagnosis and treatment program (CM-DTP) is mentioned in the introduction in the last paragraph from line 84-96 to provide the background and context of this long-term study. Details of this published program are described in reference 9, which is now cited. (page 4)

Also, Also, we provide details for what the CM-DTP entails in the sub-heading (CM care in the CM-DTP Group) line 128. (Page 6)

5. Line 108: You have mentioned the tests that were not routinely available, can you mention which tests were being used and the ones you are referring to in line 112? This is not clear in your text.

Response: 

To improve clarity, we changed this text to: “In the Historical Group, doctors relied on clinical judgment, and available tests including rapid lateral flow assays for CM diagnosis, gram stain and culture, and blood chemistry for monitoring CM patients were usually not performed because they were unavailable in the hospital or were too expensive at outside commercial laboratories for patients to afford.

6. Line 106: You mention 103 patients in historical group but line 114 mentions 105??

Response: This is corrected, the correct figure is 105, with 103 having data on ART use captured. We have clarified this in the re-submission which also matches what is in table 1. (line 108, Page 5)

7. Verify number of patients in historical group

Response: This has been corrected as above, 105 patients, (Page 5)

8. Line 211: Origin of this denominator 330 is not clear.

Response: This is denominator is changed to 318 to match the findings reported in table 1 and excludes 12 patients for whom no data was available. (Ref Table 1, line 212-214, and data line 107 page 5)

9. Table 1: The last row of patients who were untraceable; can you provide more information on these. Are they a part of the missing data for all the different variables?

Response: These CM patients were initially treated at the hospital, but no further information on their survival was not available, especially for the historical cohort at the time of evaluation because outpatient follow-up was not routine practice at the time. Subsequent analyses were not conducted on untraceable patients; therefore, they are not considered missing data for the historical group, as no further analyses were conducted for them. For the CM-DTP group, this were the missing data group (6.6%) which we mentioned in the statistical analysis reported in line 194.

(Ref: Line 228 – 237, page 12, and Line 194 page 9)

10. Line 228: This is supposed to be reporting results from your study. Factors should be discussed in the discussion.

Response: We concur with the reviewer's suggestion that this section should primarily report results. We removed the sentence regarding potential reasons for poor outcomes as previously described in reference 24.

11. Table 2: Last row; what comparison is this p-value related to?

Response: We meant the “overall group” or all patients (with available data on outcomes). However, we removed it from the table and retained the description in the text. Table. 2 (page 13)

12. Table 3 results: Which is your outcome measure? Is it a risk or a rate? Results just mention hazard from lines 255-264.

Response: Our outcome measure is Hazard in the context of Cox proportional hazards regression, (the regression model used here). The term "hazard" refers to the instantaneous risk of an event (death) occurring at a specific point in time. It represents the chance of experiencing the event of interest (i.e., death) at a particular time given the subject has survived up to that point. We described this in the sub-heading statistical analyses. We also used rates/proportions/percentages for descriptive statistics.

(Ref: page 9, line 192-203)

13. Table 3: if you are looking at fluconazole adherence, shouldn’t the reference group be those who adhered? Same comment for follow-up adherence in both tables 3 and 4.

Response: We agree that considering "fluconazole adherence" as the null group is a valid approach (one way) for analyzing and reporting this association. We considered fluconazole “not used” as the null group because it would provide a more nuanced way for comparing the baseline to all the adherence groups evaluated (adhered and defaulted). We also interpreted, presented, and discussed the results using this logical flow.

14. Table 3: Cox regression; which variables are being adjusted for? Separate variables are mentioned below the table which are not shown in the model.

Response: We acknowledge the reviewers' observation that we did not present results from a single model or provide estimates for the covariates from one output alone. Our approach was intentional to avoid committing a "table two fallacy," which could potentially arise from presenting the data in such a manner. This would be a misrepresentation of our results from the multivariable analyses.

15. Is it possible that survival was impacted by other comorbidities in this patient group? These have not been mentioned. As you analyze the mortality data as all-cause, it might be better to actually categorize CM associated deaths compared to non-CM. Otherwise, improving outcomes with the suggested interventions would not help if cause of death is non-CM.

Response: We agree that there are residual confounders that are possible, which could affect the observed results, and we mentioned this in the discussion (line 367) together with other limitations which are possible in the study. However, the observed estimates are strong and are unlikely to be solely due to residual confounders. 

Also, we would like to clarify that we were unable to categorize the deaths, as CM associated versus non-CM because we did not have information to accurately differentiate between these possibilities and that is why our case definition defined all-cause mortality. Autopsy or other post-mortem investigation was not performed to conclude causes of death.

---

## [Editor Report · Decision Letter 1]

1 May 2024

Poor Long-Term Outcomes Despite Improved Hospital Survival for Patients with Cryptococcal Meningitis in Rural, Northern Uganda

PONE-D-24-11811R1

Dear Dr. Okwir,

We’re pleased to inform you that your manuscript has been judged scientifically suitable for publication and will be formally accepted for publication once it meets all outstanding technical requirements.

Kind regards,

Felix Bongomin, MB ChB, MSc, MMed, FECMM

Academic Editor

PLOS ONE
---

## [Editor Report · Acceptance letter]

9 May 2024

PONE-D-24-11811R1 

PLOS ONE

Dear Dr. Okwir, 

I'm pleased to inform you that your manuscript has been deemed suitable for publication in PLOS ONE. Congratulations! Your manuscript is now being handed over to our production team.

Kind regards, 

on behalf of

Dr. Felix Bongomin 

Academic Editor

PLOS ONE